# Selective and Concentrative Enteropancreatic Recirculation of Antibiotics by Pigs

**DOI:** 10.3390/antibiotics13010012

**Published:** 2023-12-21

**Authors:** Karyl K. Buddington, Stefan G. Pierzynowski, William E. Holmes, Randal K. Buddington

**Affiliations:** 1Department of Biology, University of Memphis, Memphis, TN 38152, USA; 2Department of Biology, Lund University, Sölvegatan 35, 22362 Lund, Sweden; stefan.pierzynowski@biol.lu.se; 3Department of Medical Biology, IMW, Jaczewskiego 2, 20-950 Lublin, Poland; 4Department of Chemical Engineering, University of Louisiana, Lafayette, LA 70503, USA; william.holmes@louisiana.edu; 5Department of Health Sciences, University of Memphis, Memphis, TN 38152, USA; 6Stonewall Research Facility, LSU Health Sciences, Stonewall, LA 71078, USA

**Keywords:** pancreas, antibiotic, secretion, enteropancreatic recirculation, exocrine, ductal epithelium, pig

## Abstract

Antibiotics that are efficacious for infectious pancreatitis are present in pancreatic exocrine secretion (PES) after intravenous administration and above minimal inhibitory concentrations. We measured concentrations of four antibiotics by tandem liquid chromatography–mass spectroscopy in plasma and PES after enteral administration to juvenile pigs with jugular catheters and re-entrant pancreatic-duodenal catheters. Nystatin, which is not absorbed by the intestine nor used for infectious pancreatitis (negative control), was not detected in plasma or PES. Concentrations of amoxicillin increased in plasma after administration (*p* = 0.035), but not in PES (*p* = 0.51). Metronidazole and enrofloxacin that are used for infectious pancreatitis increased in plasma after enteral administration and even more so in PES, with concentrations in PES averaging 3.1 (±0.5)- and 2.3 (±0.6)-fold higher than in plasma, respectively (*p*′s < 0.001). The increase in enrofloxacin in PES relative to plasma was lower after intramuscular administration (1.8 ± 0.5; *p* = 0.001). The present results demonstrate the presence of a selective and concentrative enteropancreatic pathway of secretion for some antibiotics. Unlike the regulated secretion of bile, the constitutive secretion of PES and intestinal reabsorption may provide a continuous exposure of pancreas tissue and the small intestine to recirculated antibiotics and potentially other therapeutic molecules. There is a need to better understand the enteropancreatic recirculation of antibiotics and the associated mechanisms.

## 1. Introduction

Antibiotics are an important component of therapy for treatment of pancreatitis associated with bacterial infection [1,2,3], though with some risk [4]. The selection of specific antibiotics for the treatment of infectious pancreatitis is largely based on a combination of bacterial sensitivity, penetration into pancreatic tissue after intravenous administration, and the presence in PES above minimal inhibitory concentrations [5]. More than seven decades of studies have screened numerous antibiotics and other therapeutic molecules for the ability to traverse the blood–pancreas–ductal epithelium barriers and enter the PES. Although some antibiotics can penetrate and accumulate in pancreas tissue, this may not coincide with transfer into the PES, apparently because of a ductal epithelial barrier that limits the transfer into the PES [6].

The recirculation of bile acids involving intestinal absorption, liver absorption from the blood, and secretion into the bile for return to the intestine (enterohepatic recirculation) is well recognized. The regulated secretion of bile is associated with secondary peaks in systemic concentrations and long terminal half-lives of elimination for some therapeutic compounds following oral administration [7,8,9]. The possibility of enteropancreatic recirculation (Figure 1) has not been directly evaluated. Suggestive evidence of enteropancreatic recirculation comes from antibiotics, such as fluoroquinolones and metronidazole, that are available in oral dosage forms because of intestinal absorption and appear in PES after intravenous administration [10,11,12,13] indicative of crossing the blood-pancreas-ductal epithelium barriers. Unlike the regulated secretion of bile, the constitutive component of PES, which includes continuous secretion of antimicrobial peptides [14,15,16], would provide a more consistent exposure of pancreas tissue, notably the ductal epithelium, and the small intestine to recirculated antibiotics and other therapeutic molecules.

The objective of the present study was to elucidate the presence and specificity of a pathway for enteropancreatic recirculation (Figure 1) by comparing the appearance of four antibiotics in plasma and PES after direct administration into the duodenum. Understanding the presence of enteropancreatic recirculation and associated exposure of the pancreatic ductal epithelium to antibiotics will enhance efforts to develop more effective antibiotics for treating infections associated with pancreatitis and potentially therapeutics for pancreatic ductal adenocarcinomas, the most common form of pancreatic cancer. 

The four antibiotics selected (amoxicillin, metronidazole, enrofloxacin, nystatin) differ with respect to systemic availability after oral administration and reports of penetration into pancreatic tissue and secretion in PES when administered systemically. Metronidazole and the fluoroquinolone enrofloxacin penetrate into pancreatic tissue and appear in PES after intravenous administration and are used for bacterial associated pancreatitis [3,12,17]. Amoxicillin is absorbed by the intestine [18], but accumulation in pancreatic tissue is low compared to other antibiotics used for pancreatitis [19]. Nystatin is not absorbed appreciably by the intestine [20] and was selected as a negative control.

Pigs were selected as the model species because of translational relevance for gastroenterology physiology [21]. Importantly, the pancreatic duct of pigs is separate from the bile duct, allowing for serial collections of PES that is not contaminated by bile, which can be a significant pathway of antibiotic secretion into the intestine with higher concentrations of some antibiotics in bile than serum and PES [22]. The comparisons used healthy juvenile pigs with jugular catheters for collection of blood and re-entrant duodenum–pancreatic catheter systems [23] for direct administration of the antibiotics into the duodenum and for the collection of uncontaminated PES.

## 2. Results

The direct measurement of each antibiotic was performed using tandem liquid chromatography–mass spectroscopy. This provides a more accurate assessment than measuring antibacterial activity by the agar well diffusion method, which would include the endogenous antimicrobial activity of the PES [14,24] that is known to be present in pigs [15,16]. 

**Nystatin.** The selection of nystatin as a negative control for enteropancreatic recirculation was validated. Concentrations of nystatin in plasma after duodenal administration (0.015 μg/mL ± 0.030 for all post administration times; Figure 2) did not differ from background levels measured in the plasma of pigs receiving amoxicillin or enrofloxacin (*p* = 0.26). Nystatin in PES (0.015 μg/mL ± 0.034) did not exceed values measured in corresponding plasma samples (*p* = 0.84) or background levels in PES collected from pigs administered amoxicillin or enrofloxacin (*p* = 0.42). Hence, direct administration of nystatin into the duodenum did not result in a significant increase in plasma or PES concentrations relative to background levels.

**Metronidazole.** The highest concentrations of metronidazole were measured in samples of plasma and PES collected 20 min after duodenal administration (5.42 μg/mL ± 0.36 and 17.40 ± 1.30, respectively). Concentrations declined thereafter to 6 h (Figure 3A). Throughout the 6 h collection period, the concentrations measured in the PES exceeded those in plasma by an average of 3.1-fold (Figure 3B; *p* < 0.05). Hence, metronidazole is an example of an antibiotic that is rapidly absorbed by the intestine, appears in plasma, and is subsequently secreted in PES at concentrations that exceed those in plasma.

**Enrofloxacin.** Both routes of administration for enrofloxacin (duodenal and IM) increased concentrations in the plasma and PES (Figure 4A,B). Despite an IM dose that was 50% of the duodenal dose, plasma concentrations averaged 48% higher (±11% for all eight post-administration sampling times; *p* = 0.045) compared with administration into the duodenum. However, PES concentrations were not higher after IM administration compared with the duodenal dose (*p* = 0.63 for pooled data for all eight time points). Duodenal administration of enrofloxacin resulted in higher PES concentration during the first 2 h (Figure 4C). Over the entire 6 h collection period, ratios calculated from concentrations in PES relative to plasma averaged higher for the larger duodenal dose relative to intramuscular administration (2.32 ± 0.57 vs. 1.79 ± 0.53; *p* < 0.001).

**Amoxicillin.** Although plasma concentrations remained low after dosing (Figure 5), there was a slight but still significant increase compared with before dosing (*p* = 0.03). Furthermore, concentrations after dosing were higher than those measured in plasma collected from pigs after administering the other antibiotics (0.060 ± 0.092 vs. 0.023 ± 0.034; *p* = 0.002). These data indicate a limited amount of the amoxicillin dose was absorbed by the intestine. Concentrations of amoxicillin in the PES of dosed pigs were not higher than those measured in the plasma (*p* = 0.096) or in PES collected before administering the amoxicillin (*p* = 0.10) or the other antibiotics (*p* = 0.22). These findings suggest the pancreas has minimal ability to secrete amoxicillin in PES.

**Glucose.** Plasma concentrations of glucose were within the normal range of 100–120 mg/dL [25]. Glucose was not detected in the PES, implying the barrier between the blood and PES is selective.

## 3. Discussion

The use of healthy juvenile pigs avoids the confounding factors that complicate interpretations of data from human subjects being treated for pancreatitis, pancreatic cancer, necrotizing pancreatitis, or other pathologies that can disrupt the integrity of the pancreas and the ductal epithelial barrier and thereby affect penetration of antibiotics and other drugs into the pancreas and subsequent appearance into the PES. For example, patients with pancreatic fistulas have higher ciprofloxacin concentrations in bile and PES compared with serum 3 h after oral administration [26], whereas patients with a transplanted pancreas are unable to concentrate ciprofloxacin and pefloxacin in PES relative to serum [27,28,29]. Notably, pancreatitis alters tissue penetration [30,31], disrupts the tight junctions of the ductal epithelium [32], reduces secretion of immunoglobulin A in PES [33], thereby potentially compromises the recirculation of antibiotics.

The selective secretion of some antibiotics into PES after intravenous administration has been long recognized [10,34,35,36,37], with some antibiotics reaching concentrations in PES that are up to four-fold higher than in plasma. The higher concentrations of the chemotherapeutic agent 5-fluoruracil in PES relative to plasma [38] is another example of the capability of the pancreas ductal epithelium to selectively concentrate some therapeutic compounds in PES. The previous studies and the findings of this study confirm that the blood-pancreas-ductal epithelium pathway limits the transfer of some molecules from blood into the PES (e.g., glucose) with the ability to actively transport and concentrate others in the PES.

Metronidazole concentrations in most tissues are 60–100% of those measured in plasma [39]. Although saturable or concentrative mechanisms of transport have not been described for metronidazole, concentrative secretion of metronidazole has been reported for the gastric mucosa [40]. The two-fold higher concentrations of metronidazole measured in PES relative to plasma indicate the ductal epithelium of the pancreas has an active, carrier-mediated process capable of establishing a concentration gradient. Enrofloxacin and other fluoroquinolones have high tissue penetration [41]. The higher concentrations of enrofloxacin in PES compared with plasma indicate the basolateral and apical membranes of the ductal epithelium are capable of selective and concentrative secretion. Our findings that concentrations of metronidazole and enrofloxacin in PES after duodenal administration exceed those measured in plasma demonstrates a novel and previously unreported enteropancreatic recirculation pathway for these antibiotics and possibly other therapeutic compounds. This pathway may apply to the antibiotic trimethoprim that is absorbed after enteral administration [42] and achieves higher concentrations in PES than serum after intravenous administration [43]. Although oral doses of 5-fluoruracil cause intestinal damage [44], intestinal absorption of 5-fluoruracil [45] combined with the higher concentrations of 5-fluorouracil in PES than serum [38] are suggestive that enteropancreatic recirculation is possible for this chemotherapeutic agent. Importantly, enteropancreatic recirculation of antibiotics and other medications potentially prolongs exposure of the ductal epithelium to those antibiotics and medications and would extend the therapeutic half-life. The recirculation and concentrative secretion in PES corresponds with the recommended use of metronidazole and fluoroquinolones, such enrofloxacin and ciprofloxacin for treatment of infectious pancreatitis [3,17,46], although the efficacy has been questioned for prophylaxis [47].

The low plasma concentrations of amoxicillin measured in the pigs after duodenal administration is consistent with limited systemic bioavailability [48]. However, the low plasma appearance could have been due to competition for transporters between amoxicillin and components of the feed that was fed after administering the dose. This would explain why plasma concentrations of amoxicillin measured in the present pigs, despite a dose of 45 mg/kg, were markedly lower compared with pigs that were fasted and not fed after an oral dose of 20 mg/kg (0.06 μg/mL + 0.02 vs 3.14 µg/mL) [49,50]. Similarly, peak systemic bioavailability after providing an oral dose of 15 mg/kg amoxicillin to 25–35 kg pigs occurred before feeding [51]. The presence of a secretory pathway for amoxicillin has been speculated but ruled out for saliva and gastric juice [52]. The low PES concentrations of amoxicillin after an oral dose (present study) and for other β-lactams after intravenous administration [53,54] indicates transfer to PES is minimal, contributing to the minimal efficacy of amoxicillin for treating infectious pancreatitis.

Imipenem, a carbapenem member of beta-lactam antibiotics, is considered effective for infectious pancreatitis [55]. Like metronidazole and fluoroquinolones, imipenem is characterized by high penetration into pancreatic tissue [56,57] and secretion into PES [58]. The possibility of enteropancreatic recirculation needs to be evaluated for imipenem and other antibiotics used or considered for treatment of infectious pancreatitis.

Nystatin is not absorbed by the intestine, corresponding with the absence in plasma and PES samples, and is not used for fungal infections of the pancreas. Fluconazole is used to treat fungal infections and penetrates the pancreas and is present in PES after IV administration [59]. The availability of fluconazole after oral administration [60] suggests the possibility of enteropancreatic recirculation.

Potential mechanisms. Active concentrative secretion of bicarbonate by the ductal epithelium is well established [61] and involves activities of multiple proteins in the apical and basolateral membranes. There is interest in transporters for various solutes that are associated with the ductal epithelium as potential markers of pancreatic cancer and as therapeutic targets for more effective chemotherapeutic agents and protocols [62,63,64,65,66]. Although the mechanisms underlying the concentrative secretion by the ductal epithelium of some antibiotics into PES have not been elucidated, it is likely the secreted antibiotics share transporters identified for drugs [67,68]. Pancreatitis changes the expression of multiple solute transporters [69], and selective blocking of some solute transporters can exacerbate pancreatitis [70].

Intestinal absorption of fluoroquinolones is via organic anion transporters (OATs) in the apical membrane of enterocytes [71], whereas in breast tissue and liver enrofloxacin and other fluoroquinolones are substrates for ABC efflux transporters, corresponding with secretion of fluoroquinolones in milk and bile [72]. Saturable transport of the related ciprofloxacin by fibroblasts is responsible for secretion in gingival fluid [73], with concentrations up to four-fold higher than in plasma. The potential polarization of absorptive and secretory functions in the basolateral and apical membranes, respectively, of the ductal epithelium has not been elucidated. Although many oral and intravenous antibiotics appear in bile [74], fluoroquinolones, including enrofloxacin, do not achieve elevated concentrations in bile of rabbits [75]. This implies mechanisms of transepithelial movement differ for the selective transfer of some antibiotics into bile and PES.

Intestinal absorption of amoxicillin is by a combination of carrier-mediated (PepT1) and carrier-independent processes [76] and is considered to be saturable [77]. The lack of amoxicillin secretion in PES indicates either the absorptive or secretory mechanism necessary for secretion, or both, are absent in the ductal epithelium.

There is long-standing interest in understanding the transfer of antibiotics across the blood-brain and blood–cerebrospinal fluid barriers for the treatment of central nervous system infections [78,79,80]. These barriers involve another epithelium characterized by selective transfer of antibiotics via diffusion and energy-dependent transporters. Similar to the ductal epithelium of the pancreas, metronidazole and fluoroquinolones penetrate the BBB and achieve therapeutic levels in the cerebrospinal fluid, corresponding with transepithelial transfer from blood to cerebrospinal fluid [81]. The use of meropenem and vancomycin for CNS infections reflects a similar requirement to cross the BBB [79].

Limitations of the present study. The use of highly specific LC-MS ensured that the measured concentrations of the antibiotics in serum and PES represent the parent compounds and did not include the potential metabolites that possess antimicrobial activities. Hence, the antibiotic levels measured in PES may underestimate actual antimicrobial activity contributed by metabolites. Metronidazole is a prodrug that is bio transformed to multiple metabolites that are eliminated mainly in the urine [82,83]. Similarly, amoxicillin is subject to biotransformed [84,85] and eliminated after oral or systemic delivery to pigs [49]. Enrofloxacin is converted to ciprofloxacin, apparently in the liver [86,87], that can penetrate into pancreas tissue and PES after intravenous administration [11]. If ciprofloxacin, like enrofloxacin, is concentrated across the blood-pancreas ductal epithelium barrier, the combined concentrations of the parent compound and metabolites in PES could enhance the therapeutic dose in PES after intravenous, intramuscular, or oral administration of enrofloxacin. Interestingly, in the present study administration of enrofloxacin into the duodenum resulted in similar PES concentrations despite lower plasma concentrations compared to intramuscular injection. It is unknown if this reflects a maximum capacity to concentrate enrofloxacin in PES.

Differences exist among species for the secretion of antibiotics and other therapeutics into PES [88] and health states of the pancreas [89]. The species differences may reflect different anatomic arrangements, such as the common biliopancreatic duct in the rat versus separate biliary and pancreatic ducts with separate papillae in pigs and dogs. This is important as concentrations of some antibiotics in bile can exceed those in serum and PES, suggesting different concentrative capabilities exist for the pancreas and bile [22].

## 4. Conclusions

The ability of an antibiotic to penetrate pancreatic tissue is one determinant of efficacy for treatment of pancreatitis [56,90,91]. However, not all antibiotics that penetrate pancreas tissue enter the PES, suggesting there is an additional barrier associated with the ductal epithelium [6,22]. The present findings provide evidence that the ductal epithelium has selective mechanisms whereby some antibiotics that are absorbed by the intestine are transferred across the blood-pancreas-ductal epithelium barriers into PES at concentrations that can exceed those in plasma. The reabsorption of these antibiotics and associated antimicrobial metabolites by the intestine and re-secretion in PES potentially extends the therapeutic exposure of the infected pancreas. The enteropancreatic recirculation of enrofloxacin and metronidazole reported herein are consistent with the recommendation of the American Gastroenterological Association for the use of these two antibiotics for treating pancreatitis with concurrent infection [17]. The mechanisms responsible for the selective and concentrative transfer of antibiotics and other molecules from blood to PES remain mostly unknown. A better understanding will provide opportunities for improving the transfer of therapeutic compounds into the PES for treatment of infectious pancreatitis or for the retention of chemotherapeutics in ductal cells to treat pancreatic adenocarcinoma, as well as improving treatments for other diseases of the pancreas.

## 5. Materials and Methods

**Animals.** The studies reported herein used six healthy and recently weaned pigs obtained from a Swedish Landrace herd (Odarslöv’s Research Farm, Swedish University of Agricultural Sciences, Lund) with an average BW of 13.6 kg at the beginning of the experiment. The pigs were housed individually in pens (1 × 2 m), had free access to water, and were kept under 12 h light/12 h dark cycles (light on from 08.00 h to 20.00 h).

**Surgical procedures**. The pigs were sedated with azaperone (Stresnil, Janssen Pharmaceutica, Beerse, Belgium; 2 mg/kg BW), and anesthetization was induced with Halothane (ISC Chemicals Ltd., London, UK). Surgery was performed under aseptic conditions. The pigs were surgically fitted with a modification of a chronic pancreatic duct catheter with a T-shaped duodenal cannula for the collection and subsequent return of pancreatic juice into the duodenum [23,92]. The pigs were allowed to recover for one week after the surgery before the study was started. The pigs were not treated with antibiotics after surgery.

**Selection and administration of antibiotics and collection of samples**. Metronidazole (Flagyl^®^; MW = 171 g/mol) was selected as a candidate for enteropancreatic recirculation because of its high oral bioavailability [39], known penetration into pancreas tissue and appearance in PES after IV administration [10,13], and reported efficacy for treating anaerobic infections associated with pancreatitis [2]. In contrast, Nystatin (MW = 926) is poorly absorbed across mucous membranes and was selected as a negative control for intestinal absorption. The expectation was that after administration of Nystatin into the duodenum, concentrations would be negligible in plasma and PES. Metronidazole and Nystatin were co-administered via the duodenal catheter at dosages of 25 mg/kg and 40 mg/kg (200,000 units/kg), respectively.

Amoxicillin is absorbed by the intestine better than other β-lactam antibiotics but has low penetration into pancreatic tissue [18]. It was anticipated that after administering 45 mg/kg via the duodenal catheter, amoxicillin would be detected in the blood, but concentrations in the PES would be low or undetectable.

Enrofloxacin (Baytril^®^; MW 359 g/mol) is a fluoroquinolone used orally and intravenously for infections in animals. The efficacy of the metabolite ciprofloxacin used clinically as a prophylactic for acute necrotizing pancreatitis has been questioned, since bacterial infection is often not a contributing cause of morbidity in acute pancreatitis [93]. Nonetheless, fluoroquinolones penetrate pancreas tissue [5] and can effectively treat bacterial infections of the pancreas [2]. Ciprofloxacin has been detected in the intestinal contents of pigs after intramuscular injection [94], indicating that pancreatic secretion contributes to its elimination, but the relative contributions of biliary, pancreatic, and intestinal secretion were not determined. Enrofloxacin has been found in bile [86], but its pharmacokinetics are not affected by liver damage [87], suggesting there may be an alternative route for the recirculation of enrofloxacin, specifically via the PES. A suspension of Enrofloxacin (Baytril^®^; Elanco Animal Health, Greenfield, IN, USA) was prepared by pulverizing tablets, dissolving the powder in water, and injected via the duodenal catheter to provide a dose of 5 mg/kg. In addition, a commercially available injectable suspension of 22.7 mg/mL Enrofloxacin was injected intramuscularly (IM) at a dose of 2.5 mg/kg. The dose was administered IM, not intravenously, to avoid the contamination of the jugular catheter with Enrofloxacin.

The antibiotics were administered between 0800 and 0900 prior to feeding. Each antibiotic was administered to three pigs, and each pig was used to study two antibiotics, with a minimum of 3 intervening days between the administration of the first and second antibiotics. Collections of PES and blood (with EDTA for isolation of plasma) were made immediately before administration to obtain baseline values and at 20, 40, and 60 min after administration and hourly thereafter to 6 h. The pigs were provided food after administration of the antibiotic(s). The PES and plasma samples were frozen (−80 °C) until analyzed.

**Analytical procedures**. Each of the collected plasma and PES samples were analyzed for all four antibiotics. PES was diluted 1:9 in 1M Tris Buffer (pH 7.4) and plasma in acetonitrile (1:9). After centrifugation (13,000× *g*; 10 min, 4 °C), the supernatants were filtered (0.45 μm PTFE filter), separated using RP-HPLC (Restek Allure PFP Propyl 3 μm 100 × 2.1 mm column using Varian ProStar 210 solvent delivery system at 0.2 mL/min), and analyzed by MS/MS (Bruker Esquire; 4.5 kV ionization energy, positive ionization, 30 PSI nebulizer, 10.0 L/min dry gas and 325 °C). Each run included a standard curve with antibiotic concentrations (ng/mL) of 3.13, 6.25, 12.5, 25.0, and 50. A buffer blank was included before each time series for a pig. Spiked samples of either PES or serum corresponding with sample type were included at the end of the time series to evaluate accuracy, reproducibility, and recovery. The 95% detection limit confidences were established for each antibiotic.

Glucose was measured using a commercial kit in 5 concurrently collected pairs of PES and plasma samples. This was done to obtain additional insights about the selectivity of the appearance of the antibiotics and glucose in the PES.

## Figures and Tables

**Figure 1 antibiotics-13-00012-f001:**
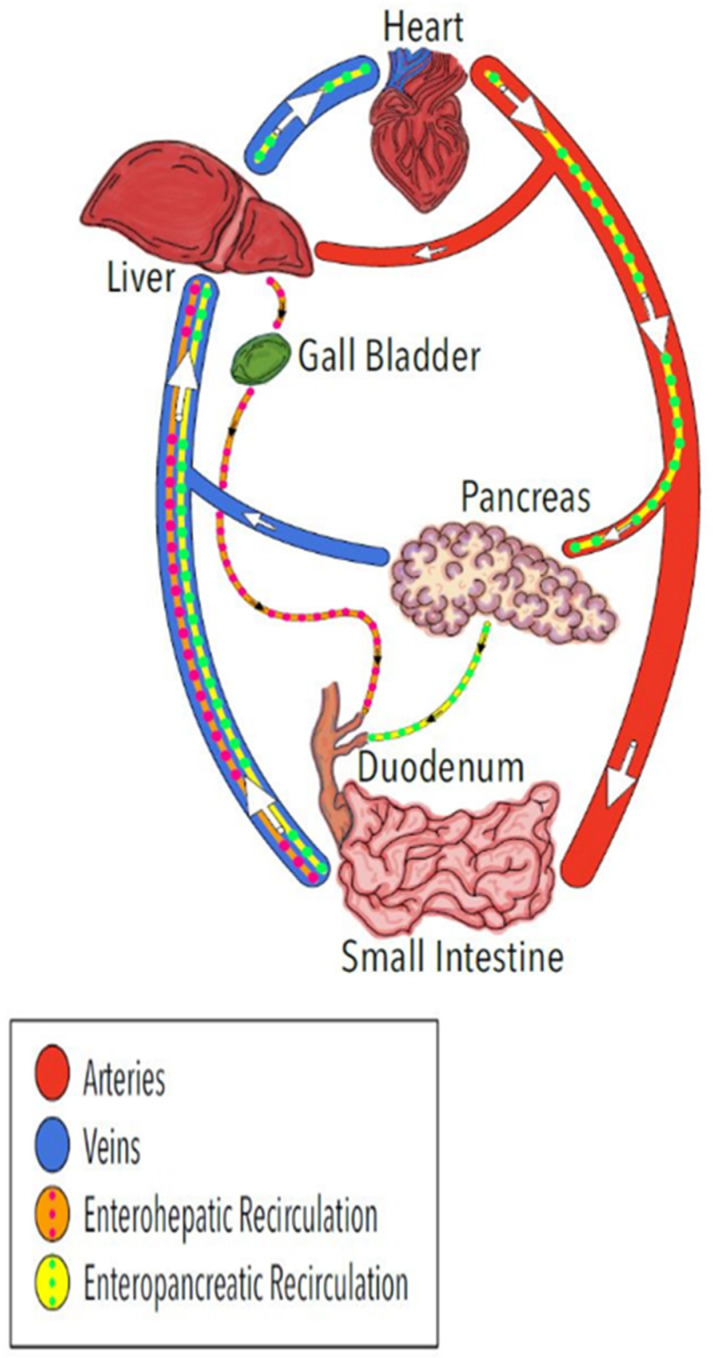
Illustration of the well-known enterohepatic recirculation pathway for bile acids and other molecules (orange) and the proposed enteropancreatic recirculation of some antibiotics and possibly other molecules (yellow). The arrows indicate direction of blood flow in the veins and arteries.

**Figure 2 antibiotics-13-00012-f002:**
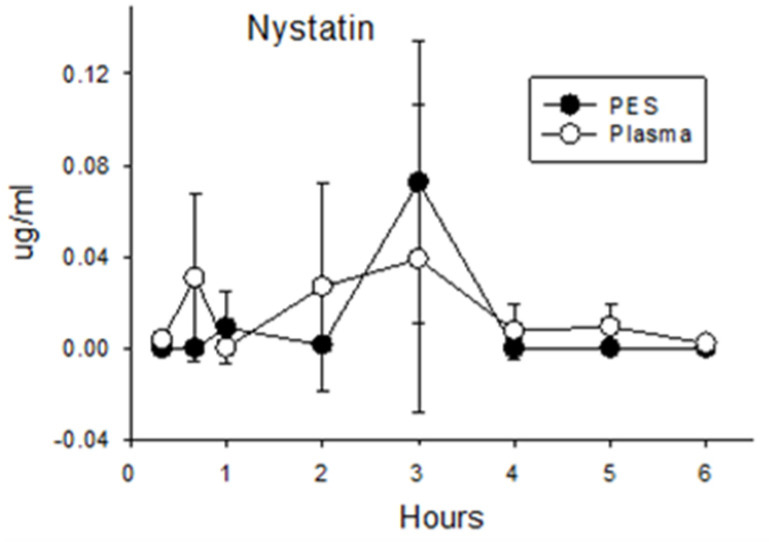
Concentrations of nystatin in plasma and PES after administration of 40 mg/kg (200,000 units/kg) directly into the duodenum.

**Figure 3 antibiotics-13-00012-f003:**
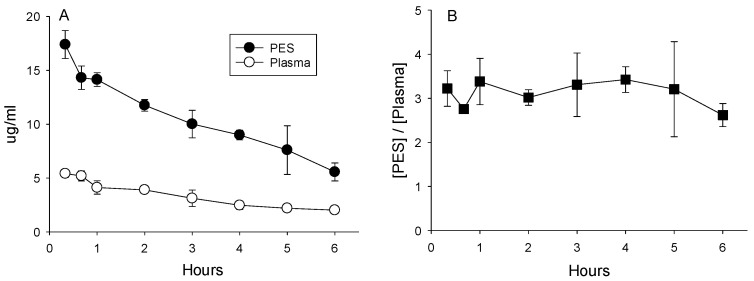
(**A**) Concentrations of metronidazole in plasma and PES after administration of 25 mg/kg directly into the duodenum. (**B**) Ratios calculated as the quotient of concentrations of metronidazole in the PES divided by plasma averaged 3.1 ± 0.5, indicative of concentrative secretion.

**Figure 4 antibiotics-13-00012-f004:**
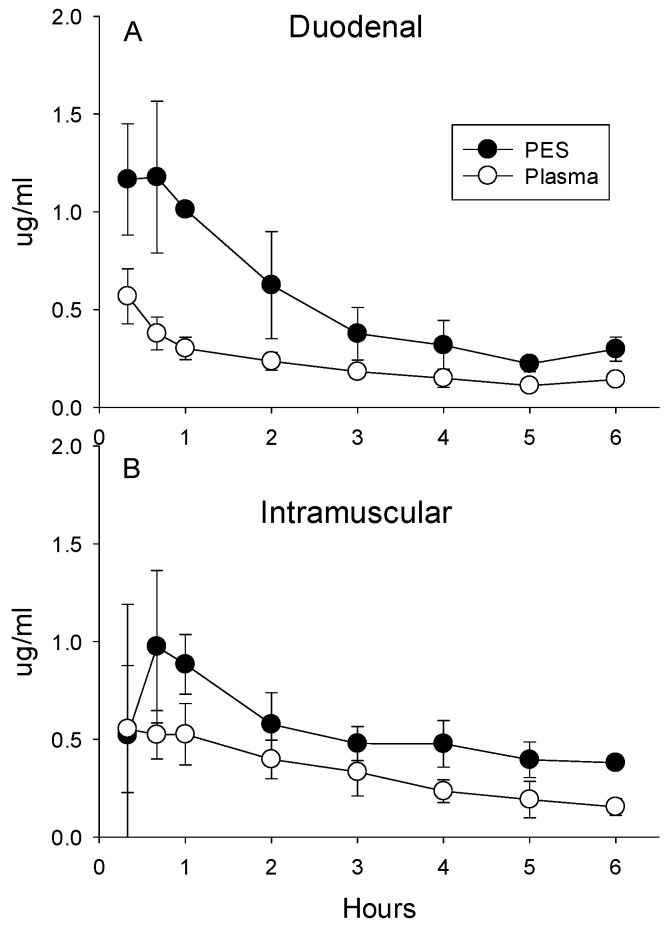
(**A**) Concentrations of enrofloxacin in plasma and PES after the administration of 5 mg/kg directly into the duodenum and (**B**) intramuscular administration of 2.5 mg/kg. (**C**) Ratios calculated as the quotient of concentrations of enrofloxacin in the PES divided by plasma after administration into the duodenum or intramuscular averaged 3.1 + 0.5, indicative of concentrative secretion. Asterisks indicate ratios differed significantly between intramuscular and duodenal routes of administration.

**Figure 5 antibiotics-13-00012-f005:**
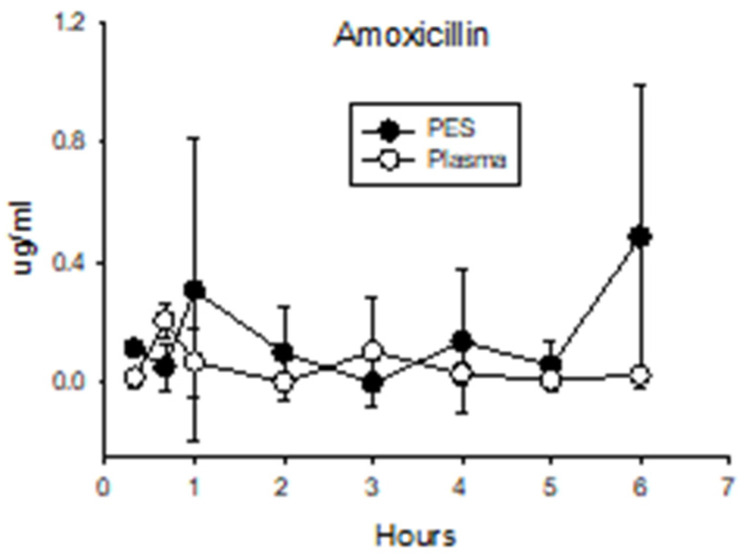
Concentrations of amoxicillin in plasma and PES after administration of 45 mg/kg directly into the duodenum.

## Data Availability

All data collected and analyzed during this study are presented in this published article.

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
