# Peer review of "Selective and Concentrative Enteropancreatic Recirculation of Antibiotics by Pigs"

_antibiotics, 2023, doi:10.3390/antibiotics13010012_

Round 1

Reviewer 1 Report

Comments and Suggestions for Authors

the manuscript needs to be thoroughly revised for improving language and clarity of understanding.

 the abstract is difficult to understand. please make clear sentences. methodology should be briefly described.

The selection criteria of the four antibiotics used should be clear.

line 134-5-Although plasma concentrations remained low after dosing (Figure 5), there was a significant increase compared to before dosing. - difficult to comprehend.

discussion should be more elaborated.

Comments on the Quality of English Language

very poor

Author Response

Responses to Reviewer 1

Quality of English Language

( ) I am not qualified to assess the quality of English in this paper

(x) English very difficult to understand/incomprehensible

( ) Extensive editing of English language required

( ) Moderate editing of English language required

( ) Minor editing of English language required

( ) English language fine. No issues detected

We are puzzled by this comment.  Three of the authors are US citizens by birth and lifelong English speakers with multiple publications in English.  We also had other English speakers review the manuscript and offer comments.  The second reviewer considered the manuscript to be “well written”.  Regardless, we have reviewed the entire manuscript for quality of English and clarified statements that might be confusing.

Yes          Can be improved              Must be improved           Not applicable

Does the introduction provide sufficient background and include all relevant references?

( )            ( )            (x)           ( )

Are all the cited references relevant to the research?

( )            (x)           ( )            ( )

Is the research design appropriate?

(x)           ( )            ( )            ( )

Are the methods adequately described?

( )            ( )            (x)           ( )

Are the results clearly presented?

( )            ( )            (x)           ( )

Are the conclusions supported by the results?

( )            (x)           ( )            ( )

Comments and Suggestions for Authors

the manuscript needs to be thoroughly revised for improving language and clarity of understanding. 

As described above, we are puzzled by the comment about poor English.  We have revised the introduction and discussion extensively to improve clarity and include additional information and references.

 the abstract is difficult to understand. please make clear sentences. methodology should be briefly described.

We include in the abstract how the four antibiotics were measured using tandem liquid chromatography – mass spectroscopy.  Details are provided in the methods section about how the four antibiotics were measured and quantified in plasma and PES using LC-MS.

The selection criteria of the four antibiotics used should be clear.

We have expanded the description of why we selected the four antibiotics.

line 134-5-Although plasma concentrations remained low after dosing (Figure 5), there was a significant increase compared to before dosing. - difficult to comprehend.

We have included in the sentence how the increase of amoxicillin in plasma after dosing was slight, but still significant.

discussion should be more elaborated.

We have extensively revised and expanded the discussion to elaborate on potential mechanisms for the transfer of some but not all antibiotics from the blood to the PES.  We also mention limitations of our study.

Reviewer 2 Report

Comments and Suggestions for Authors

This the review of the article “Selective and concentrative enteropancreatic recirculation of antibiotics by pigs”.

The article is well written and its experiments well designed.

Some points needs to be amended or clarified.

Line 49, the article used a reference (8) doesn’t talk about benzyl-pyrimidines, I think need to be changed.

Line 51-53, I cannot follow authors claims that “these findings suggest” what they are writing about. Needs to be re-written.

Line 66-70, authors discuss about antibiotics and other cancer therapeutics like it’s the same pharmacokinetics and pharmacodynamics. Needs to me omitted or re-written. Also, this paragraph is not supported by any references.

Line 187, the secretion of Enrofloxacin in bile, milk is due to the same epithelium? Also the reference (42) used is talking about metronidazole and not Enrofloxacin.

Also there is a lack of up to date and newer articles used as references.  The authors should make an effort to use references closer to today. With the worse paragraph being the conclusion, where authors use references 30 and mostly nearly 40 years old.

Author Response

(x) English language fine. No issues detected

Yes          Can be improved              Must be improved           Not applicable

Does the introduction provide sufficient background and include all relevant references?

( )            (x)           ( )            ( )

Are all the cited references relevant to the research?

( )            ( )            (x)           ( )

Is the research design appropriate?

(x)           ( )            ( )            ( )

Are the methods adequately described?

(x)           ( )            ( )            ( )

Are the results clearly presented?

(x)           ( )            ( )            ( )

Are the conclusions supported by the results?

(x)           ( )            ( )            ( )

Comments and Suggestions for Authors

The article is well written and its experiments well designed.

Thank you.

Some points needs to be amended or clarified.

Line 49, the article used a reference (8) doesn’t talk about benzyl-pyrimidines, I think need to be changed.

We thank the reviewer for catching this mistake.  We moved the mention of benzyl pyrimidines to the discussion section where we refer to trimethoprim. 

Line 51-53, I cannot follow authors claims that “these findings suggest” what they are writing about. Needs to be re-written.

The introduction has been extensively revised.  This statement has been removed for clarification.

Line 66-70, authors discuss about antibiotics and other cancer therapeutics like it’s the same pharmacokinetics and pharmacodynamics. Needs to me omitted or re-written. Also, this paragraph is not supported by any references.

The reviewer’s comment led us to revise and expand the discussion to include a section that reviews mechanisms of drug secretion that involve various transporters that are likely to be capable of transporting some antibiotics, allowing for the transfer from the blood to the PES. 

Line 187, the secretion of Enrofloxacin in bile, milk is due to the same epithelium? Also the reference (42) used is talking about metronidazole and not Enrofloxacin.

We thank the reviewer for bringing these issues to our attention.  This section has been revised and the reference corrected.

Also there is a lack of up to date and newer articles used as references.  The authors should make an effort to use references closer to today. With the worse paragraph being the conclusion, where authors use references 30 and mostly nearly 40 years old.

We agree that many references are “dated”.  We have made an effort to include more recent references.  These include four references published in “Antibiotics” that are relevant to our findings.  Although our findings for enteropancreatic recirculation of antibiotics are unique, they are founded on older papers that provided a foundation of knowledge about the presence of antibiotics in PES after IV administration.  For any number of reasons, the findings reported 30 and more years back didn’t stimulate more recent studies about transfer of antibiotics from the blood to the PES.  As a result, there are few papers that are recent (<10 years).  We extensively reviewed the literature to find those that have been published.  Hopefully, our contribution will stimulate research that will expand the modern literature. 

Round 2

Reviewer 1 Report

Comments and Suggestions for Authors

the manuscript is fine and the comments have been incorporated.